# Jetting Performance of Polyethylene Glycol and Reactive Dye Solutions

**DOI:** 10.3390/polym11040739

**Published:** 2019-04-24

**Authors:** Zhiyuan Tang, Kuanjun Fang, Yawei Song, Fuyun Sun

**Affiliations:** 1Fiber Materials and Modern Textiles of the Growing Base for State Key Laboratory, Qingdao University, 308 Ningxia Road, Qingdao 266071, China; 18363995943@163.com (Z.T.); 15864737168@163.com (Y.S.); 18354261812@163.com (F.S.); 2School of Textiles & Clothing, Qingdao University, 308 Ningxia Road, Qingdao 266071, China; 3Collaborative Innovation Center for Eco-Textiles of Shandong Province, 308 Ningxia Road, Qingdao 266071, China

**Keywords:** jetting performance, polyethylene glycol, reactive dye, rheological property, surface tension, drop formation

## Abstract

The jetting performance of dye inks determines the image quality, production efficiency, and lifetime of the print head. In the present study, we explored the jetting performance of mixed solutions of polyethylene glycol (PEG) and reactive dye by testing the visible absorption spectra, rheological properties, and surface tension, in addition to the observation of droplet formation. The results indicate that PEG macromolecules could change the aggregate groups of Red 218 molecules into smaller ones through hydrophobic interactions and separation effect. The addition of PEG into the dye solution increased the viscosity and decreased the surface tension. In the whole shear rate range tested, the 10% and 20% PEG400, as well as the 30% PEG200 dye solutions, showed good Newtonian fluid behavior. PEG macromolecules improved the droplet formation of the dye solutions. Increasing the PEG400 concentration to 30% and 40% resulted in elimination of the formation of satellites and the formation of ideal droplets at 10,000 Hz jetting frequency. A 30% PEG600-dye solution with the *Z* value of 4.6 formed the best spherical droplets at 10,000 Hz and produced perfect color images on cotton fabrics.

## 1. Introduction

Digital inkjet printing has become an important technology in the textile industry [1,2,3,4,5,6,7,8,9]. Reactive dye inks are the most important material in determining the printed image quality of cellulose and protein fabrics [2,3,7,10,11]. However, there are still some major challenges to be faced in the actual production process. For example, satellite droplets form more easily during inkjet printing with high-drop-velocity nozzles such as Kyocera’s [12,13,14], leading to low-quality images on the fabrics. In spite of advances in ink chemistry and physics, reactive dye inks are often clogged when seasons change due to the crystallization of dissolved dyes. During textile inkjet printing, unpleasant odors are produced because some relatively volatile solvents (e.g., ethylene glycol, 2-pyrrolidone) are present in the dye inks. It has been noticed that there are marked differences in the jetting performance between batches or between colors of a set of apparently identical reactive dye inks [15]. Hence, extensive and time-consuming ink reformulation and validation need to be carried out for each batch or between colors to achieve satisfactory jetting performance. Therefore, it is of significance to develop high-performance reactive dye inks for the textile industry.

In order to avoid the satellites, other than designing new nozzle geometries, various aqueous solutions of glycerol, glycols, and polyethylene oxide have been well studied in the past years [16,17]. For example, increasing the molecular weight of polyethylene oxide resulted in decreasing the number of satellites for 50% aqueous solutions of glycerol and polyethylene oxide [17]. It has been proven that the printability of fluids is determined by the inverse (Z) of the Ohnesorge number which relates to the fluid viscosity, surface tension, and density [16,18]. In practice, dye inks seem to be more complicated than such situations. The dye/additive interaction influences the ink performance. Acid dye ink solutions with favorable physicochemical properties and low-molecular-weight poly(vinylpyrrolidone) showed good ink droplet formation [19]. Surfactant-free aqueous dispersions of poly(3,4-ethylenedioxythiophene:polystyrenesulphonate were successfully used in inkjet printing with no satellite droplets [12]. Large dye aggregates easily form dye particles when the ambient temperature decreases, inducing nozzle clogging. Poly(vinylpyrrolidone) was able to prevent the dye from forming large aggregates [20], but its high molecular weight resulted in nozzle clogging through macromolecular chain entanglement [19].

Reactive dye inks are usually composed of humectants, surfactants, solubilizers, bactericides, reactive dyes, and water [21,22]. In order to obtain deep and bright colors jetted on fabrics, it is necessary to increase the dye concentration and to decrease the other additives as much as possible. However, inks with high dye concentration need more organic solvents, which results in serious nozzle clogging and much more unpleasant odors. Therefore, understanding the dye–additive interaction in inks to increase dye concentrations, suppress odors, and improve ink stability is important in the formulation of high-performance reactive dye inks. In the present study, non-toxic and odorless PEGs which are miscible with water were used to explore their effect on dye aggregation in aqueous solutions [23,24]. We tested the physical properties, droplet formation, and the printability of mixed solutions of PEG and reactive dye in order to obtain high-performance reactive dye inks which are environmentally friendly.

## 2. Materials and Methods

### 2.1. Materials

Polyethylene glycols with different number averaged molecular weights—200 g/mol (PEG200), 400 g/mol (PEG400), and 600 g/mol (PEG600)—were purchased from Sinopharm Chemical Reagent Co., Ltd. (Shanghai, China). Pure C. I. Reactive Red 218 (Red 218) for ink preparation was purchased from Yongguang Chemical Industry Co., Ltd. (Taipei, China), and was used without further purification. The chemical structures of the Red 218 and polyethylene glycols are shown in Figure 1. All water used was pure, with a conductivity of 0.9 µS/cm. Cotton fabric (60 × 60/200 × 98) was provided by YuYue Home Textiles Co., Ltd. (Binzhou, China).

### 2.2. Preparation of PEG–Dye Mixed Solutions

The weighted reactive dye was slowly dissolved in a mixture of water and PEG under stirring by a HJ-6A digital multi-head magnetic stirrer (Shanghai Shuangjie Experimental Equipment Co., Ltd., China) at 25 °C for two hours. The mixtures of PEG and water were prepared before use. The dye concentration was 7% by weight, unless otherwise indicated. The concentrations of PEG400 in PEG–dye solutions were 0.0%, 10.0%, 20.0%, 30.0%, and 40.0% by weight, respectively, and the concentrations of PEG200 and PEG600 were 30.0%.

### 2.3. Measurement of Visible Absorption Spectra

The visible absorption spectra of the Red 218 inks with various PEG concentrations were measured using a U-3900H ultraviolet spectrophotometer (Hitachi High-Tech Co., Ltd., Tokyo, Japan) with a 0.01 mm thickness cuvette at 25 °C.

### 2.4. Test of Rheological Property and Surface Tensions

The variation of Red 218 and PEG solution viscosity vs. relatively high shear rate was measured using a FLUDICAM RHEO microfluidic visual rheometer (Formulaction company, Toulouse, France) at 25 °C. The rheological properties under low shear rates were measured using a DV-III + programmable rheometer (Brookfield company, Middleboro, America) at 25 °C. The surface tensions of Red 218 and PEG solutions were measured with a SITA surface tension meter (SITA company, Dresden, Germany) at 25 °C.

### 2.5. Density Measurement and Z Value Calculation

The densities of the PEG–dye solutions were measured in accordance with the China national standard GB/T 4472-2011 at 25 °C. The Z values of the PEG–dye solutions were calculated per Equation (1):(1)$$Z=\frac{1}{Oh}=\frac{{\left(\gamma \rho \alpha \right)}^{1/2}}{\eta },$$
where α is the diameter of the jetting nozzle, ρ is the fluid density, *η* is the fluid viscosity, and γ is the surface tension.

### 2.6. Observation of Droplet Formation and Inkjet Printing

The droplet formation in the PEG–dye solutions was observed at 25 °C using an IIA-1502 ink droplet observer (Hangzhou Fanjiang Electronic Technology Co., Ltd., China) equipped with a KJ4B300 inkjet print head (Kyocera company, Kyoto, Japan) with a nozzle diameter of 20 µm. The voltage waveform had a rise and fall time of 2 µs, the driving voltage was 26 V, and the pulse width was 6 µs. The inkjet printing was conducted with the resolution of 600 × 600 dpi at 25 °C.

## 3. Results and Discussion

### 3.1. Visible Absorption Spectra of PEG and Reactive Dye Solutions

In order to understand the dye–PEG interaction, visible absorption spectra were measured as shown in Figure 2. It is clear that there are two absorption peaks in Figure 2a for the pure Red 218 aqueous solution: one at 518 nm, the other at 554 nm. Adding 10% PEG400 in the dye solution resulted in an obvious increase of peak height, and the two peaks red-shifted to 524 and 558 nm, respectively. Increasing the PEG400 concentration to 20% led to not only obvious increases of both the peak strengths but also slight red shifts to 526 and 560 nm. Upon further increasing the PEG400 concentration to 40%, the peak strengths and the peak positions did not change evidently. This result indicates that PEG400 could make the dye aggregates become smaller through the hydrophobic interaction between the PEG and dye molecules and the separation effect of PEG macromolecules, as shown in Figure 3. Figure 2b shows that other PEGs with different molecular weights (i.e., PEG200 and PEG600) had the same effect as PEG400.

### 3.2. Rheology of the PEG–Dye Solutions

Rheological properties determine inks’ jetting ability [15,19,25,26]. The ideal inkjet inks should be Newtonian fluids. Usually, the ink rheology is characterized by apparent viscosity using rotational rheometers. Figure 4a shows that the viscosities of all PEG–dye solutions did not change with increasingly lower shear rates, indicating that the dye aqueous solutions exhibited good Newtonian fluid behavior. However, this kind of rheometer struggles to handle very low viscosities at high shear rates. Additionally, dye solutions are not necessarily Newtonian fluids. Therefore, we measured the variation of viscosity with higher shear rates close to the firing frequencies of printing heads using the FLUDICAM RHEO rheometer as shown in Figure 4b. It can be seen that adding 10% and 20% PEG400 increased the viscosity but did not change the viscosity’s variation tendency. Further increasing the PEG400 concentration to 30% resulted in slight decreases of viscosity at shear rates higher than 30,000 s^−1^. However, when the PEG400 concentration reached 40%, the viscosity began to slowly decrease at a shear rate of 5000 s^−1^. At the concentration of 30%, PEG600 had the highest viscosity compared to PEG200 and PEG400. The viscosities of 30% PEG600 and PEG400 dye solutions decreased slightly at a shear rate of 40,000 s^−1^. In the whole shear rate range, the 10% and 20% PEG400, and the 30% PEG200 dye solutions showed excellent Newtonian fluid behavior.

### 3.3. Surface Tensions of the PEG–Dye Solutions

The surface tension of ink is an important physicochemical parameter for droplet formation [19,27,28]. Stable and appropriate surface tension favors high ink jetting performance. In the present study, we used a SITA surface tension meter to measure the dynamic surface tensions of PEGs and reactive dye solutions, as shown in Figure 5.

Figure 5 shows that the surface tension of Red 218 dye aqueous solution decreased with prolonged bubble lifetime, meaning that the dye molecules arranged at the air/liquid interface as shown in Figure 6a. When PEG400 was added into the dye solutions as shown in Figure 5a, the surface tensions decreased slowly, and the values changed only slightly with increased bubble lifetime. The higher the PEG400 concentration, the lower the surface tension. Adding the same amount (30%) of PEG200, PEG400, or PEG600, the surface tensions were almost the same, meaning that the molecular weight of PEG did not affect the surface tension. This result indicates that the PEG macromolecules formed a specific structure at the air/liquid interface, as shown in Figure 6b.

Table 2 summarizes the surface tensions, densities, and viscosities of PEG–dye solutions. Based on these parameters, the Z value—a parameter for evaluating the printability of fluid—was calculated by Equation (1). It can be seen that adding PEG into the dye solution resulted in an evident decrease in the Z value. Increasing the PEG400 concentration from 10% to 40% led to a decrease in the Z value from 17.8 to 3.7. For the same concentration (30%), PEG600 had the minimum Z value of 4.6, and PEG200 had the maximum Z value of 8.3, meaning that the higher the PEG molecular weight, the smaller the Z value.

### 3.4. Droplet Formation and Inkjet Printing

Droplet formation in inkjet printing is determined by many factors, such as the jetting frequency, the driving voltage wave, the fluid viscosity, and the surface tension. Figure 7 shows the droplet formation of 7% Red 218 dye aqueous solutions at different jetting frequencies. It can be seen that at 1500 Hz the fluid was easily split into multiple droplets. Enhancing the jetting frequency could reduce the droplet numbers, and the volume and shape of satellite droplets. At 10,000 or 20,000 Hz, the shape of the jetted drop was much better than that at 1500 and 5000 Hz. Therefore, there is a proper jetting frequency for a certain fluid to form good drops.

Due to the arrangement of PEG macromolecules at the air/liquid interface, the addition of PEG in the dye solution is certain to change the formation of droplets. Figure 8 shows droplet formation images of PEG–dye solutions. It can be seen that there were still many satellite droplets in the images of 10% and 20% PEG400–dye solutions. The satellites were eliminated upon increasing the PEG400 concentration to 30% and 40%. This result indicates that sufficiently quantities of PEG400 macromolecules can make the dye and water molecules unite more tightly and reduce the surface tension of ink fluid through regular arrangement at the air/liquid interface, as shown in Figure 6b.

For 30% PEG400 and 7% dye solutions, increasing the jetting frequency from 1500 Hz to 5000 Hz led to the formation of more spherical drops (Figure 9a,b). When the frequency increased to 10,000 Hz, the drops formed spherical shapes (Figure 9c). Upon further increasing the frequency to 20,000 Hz, the formed drops became less than ideal (Figure 9d). Higher jetting frequency means a shorter time for the macromolecules to change their configurations to form spherical drops by overcoming the interaction forces of fluid molecules. Hence, there is always a proper jetting frequency for a certain PEG–dye solution.

For 30% PEG200, PEG400, and PEG600 dye solutions, the jetted droplet images are shown in Figure 10. It is clear that the PEG600–dye solution formed the best spherical drops. As Table 2 shows, the Z values for PEG200, PEG400, and PEG600 solutions were 8.3, 6.4, and 4.6, respectively. Therefore, the PEG600 solution with the Z value of 4.6 had the best printability.

According to the results, we printed cotton fabrics using the PEG600–dye solution. The result is shown in Figure 11. It is clear that 30% PEG600 solution produced an ideal inkjet printing effect with deep and bright color (Figure 11a) and with high precision (Figure 11b).

## 4. Conclusions

In aqueous solutions, PEG macromolecules could change the aggregate groups of Red 218 molecules into smaller ones through hydrophobic interactions and separation effect. The addition of PEGs into the dye solution increased the viscosity and decreased the surface tension. In the whole shear rate range tested, the 10% and 20% PEG400, and the 30% PEG200 dye solutions showed good Newtonian fluid behavior. PEG macromolecules improved the droplet formation of dye solutions. Increasing the PEG400 concentration to 30% and 40% resulted in elimination of the formation of satellites and the formation of ideal droplets at a 10,000 Hz jetting frequency. Compared with 30% PEG200 and 30% PEG400, 30% PEG600–dye solution with the Z value of 4.6 formed the best spherical drops at 10,000 Hz and produced perfect color images on cotton fabrics.

## Figures and Tables

**Figure 1 polymers-11-00739-f001:**
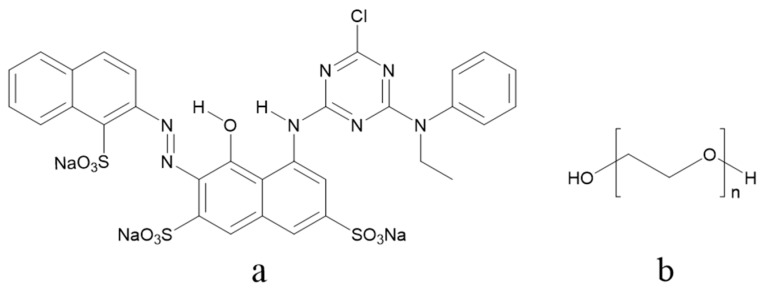
(**a**) Chemical structure of Red 218; (**b**) Chemical structure of polyethylene glycol (PEG).

**Figure 2 polymers-11-00739-f002:**
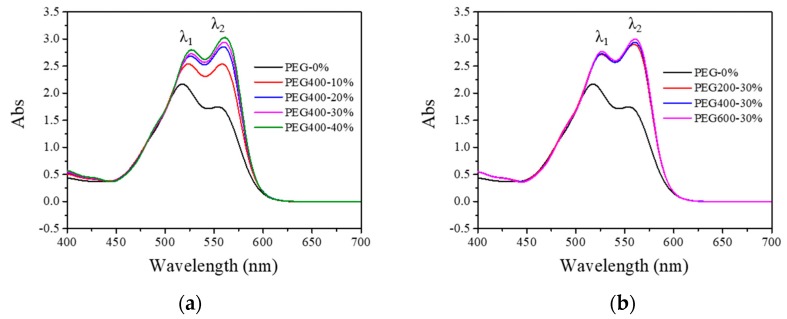
Visible absorption spectra of 7% Red 218 and PEG solutions at 25 °C. (**a**) For different amounts of PEG400; (**b**) For PEG200, PEG400, and PEG600 at 30% by weight.

**Figure 3 polymers-11-00739-f003:**
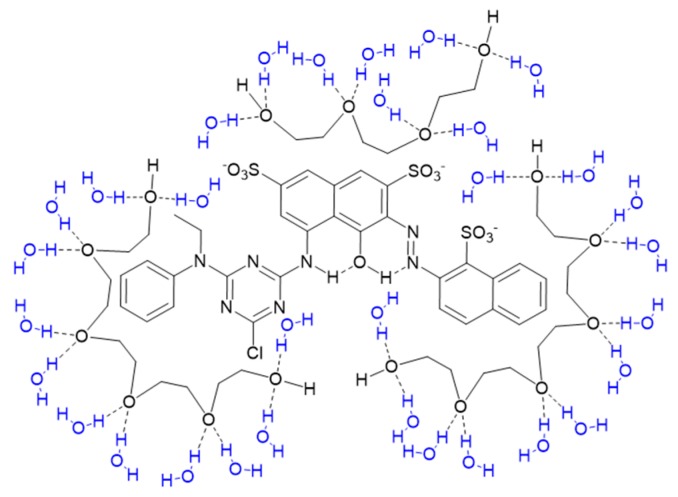
The hydrophobic interaction of PEG and dye molecules and the separation effect of PEG macromolecules. Table 1 lists all the peak positions for different dye solutions. It is clear that the maximum absorption peak of the dye solution, λ_1_, changed to λ_2_ when PEGs were added to the solution due to the rapid increase of the peak height of λ_2_.

**Figure 4 polymers-11-00739-f004:**
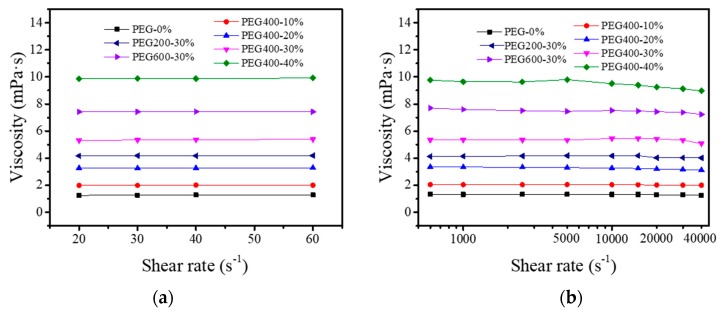
The rheological properties of Red 218 dye inks at different PEG400 concentrations and PEG molecular weights. (**a**) At low shear rates; (**b**) At high shear rates. Red 218 concentration was 7%, and the temperature was 25 °C.

**Figure 5 polymers-11-00739-f005:**
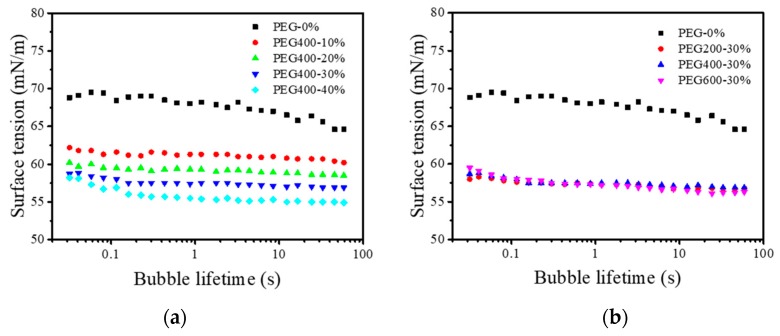
The surface tension of Red 21 8 dye inks: (**a**) At different PEG400 concentrations; (**b**) At different PEG molecular weights. The Red 218 concentration was 7%, and the temperature was 25 °C.

**Figure 6 polymers-11-00739-f006:**
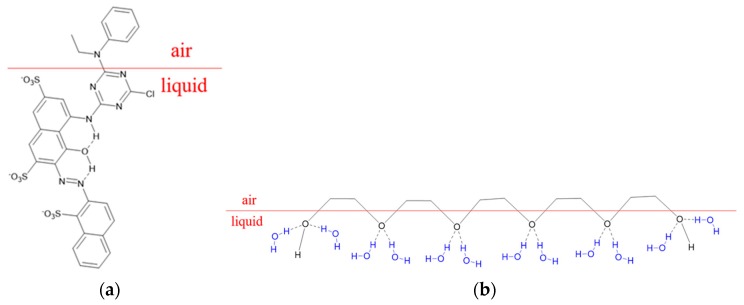
The arrangement of Red 218 dye (**a**) and PEG molecules (**b**) at the air/liquid interface.

**Figure 7 polymers-11-00739-f007:**
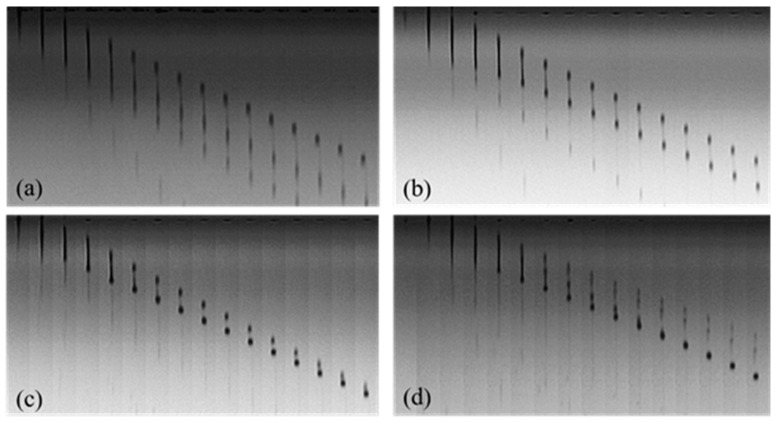
Droplet formation of 7% (*w*/*w*) Red 218 dye aqueous solutions at different jetting frequency at 25 °C. (**a**) 1500 Hz; (**b**) 5000 Hz; (**c**) 10,000 Hz; (**d**) 20,000 Hz.

**Figure 8 polymers-11-00739-f008:**
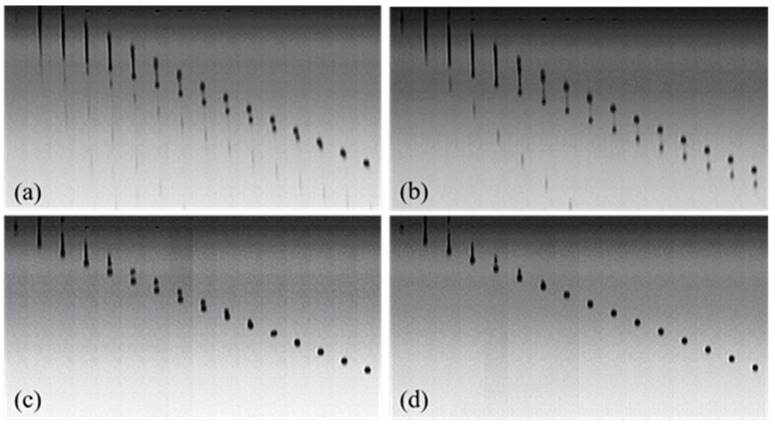
Droplet formation of 7% dye solutions with different concentrations of PEG400 at 25 °C and 10,000 Hz. (**a**) 10% PEG400, (**b**) 20% PEG400, (**c**) 30% PEG400, (**d**) 40% PEG400.

**Figure 9 polymers-11-00739-f009:**
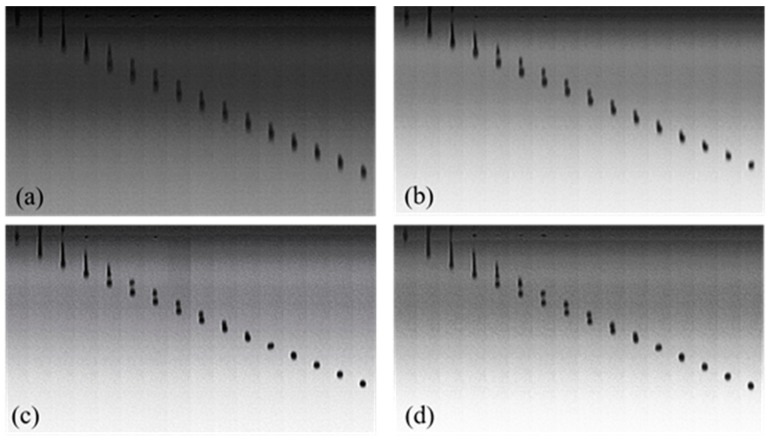
Droplet formation of 7% Red 218 dye solutions containing 30% PEG400 at different jetting frequency and 25 °C: (**a**) 1500 Hz; (**b**) 5000 Hz; (**c**) 10,000 Hz; (**d**) 20,000 Hz.

**Figure 10 polymers-11-00739-f010:**
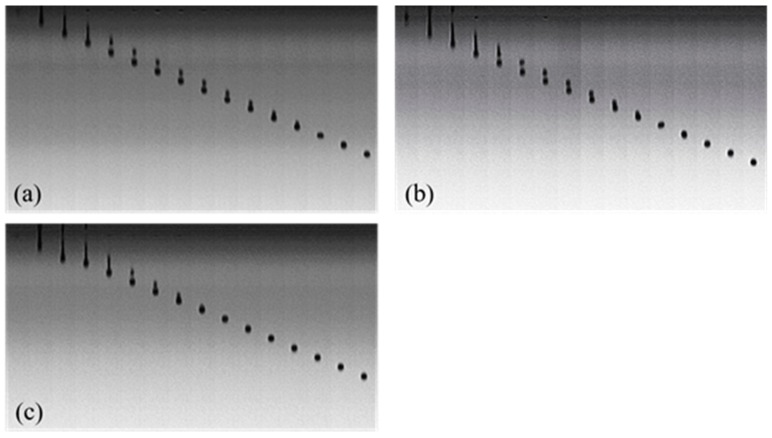
The influence of PEG molecular weights on the droplet formation of Red 218 dye solutions. (**a**) PEG200; (**b**) PEG400; (**c**) PEG600.The concentrations of Red 218 and PEG were 7% and 30%, respectively, the temperature was 25 °C, and the jet frequency was 10,000 Hz.

**Figure 11 polymers-11-00739-f011:**
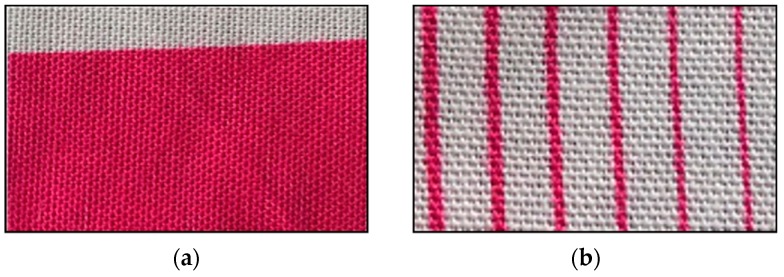
The images of cotton fabrics printed with 30% PEG600–dye solution: (**a**) The printed fabric, (**b**) The lines printed.

**Table 1 polymers-11-00739-t001:** The absorption peak positions of PEG–dye aqueous solutions.

Peak Positions	Wavelengths (nm)						
	0% PEG	10% PEG400	20% PEG400	30% PEG400	40% PEG400	30% PEG200	30% PEG600
λ_1_	518 ^a^	524	526	527	527	526	527
λ_2_	554	558 ^a^	560 ^a^	560 ^a^	561 ^a^	559 ^a^	561 ^a^

^a^ Maximum absorption wavelength.

**Table 2 polymers-11-00739-t002:** The physical properties and Z values of PEG–dye aqueous solutions.

PEG	0% PEG	10% PEG400	20% PEG400	30% PEG400	40% PEG400	30% PEG200	30% PEG600
γ ^a^ (mN/m)	65.4	60.5	58.6	57.0	55.0	56.8	56.2
ρ (kg/m^3^)	1027	1044	1061	1080	1097	1078	1080
η ^b^ (mPa·s)	1.3	2.0	3.3	5.5	9.5	4.2	7.5
Z	28.2	17.8	10.7	6.4	3.7	8.3	4.6

^a^ Average of five surface tension values; ^b^ the viscosity at a shear rate of 10,000 s^−1^.

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
