# Peer review of "Jetting Performance of Polyethylene Glycol and Reactive Dye Solutions"

_polymers, 2019, doi:10.3390/polym11040739_

Reviewer 1 Report

·         To analyse the studied problem, it is necessary to measure some quantities which determination is not generally easy. Fig. 4 can serve as such example. For better visual evaluation of the measured data an additional vertical axis on the right side would be really beneficial. The lower line should represent water viscosity at 25 oC. According to the tabulated data this value should be 0.89 mPa.s. However, a value depicted here seems to be by at least 50 % higher. On the other hand viscosity at 25 oC for pure PEG 400 is according to the literature about 88 mPa.s. A value in Fig.4 for PEG400-40% is approximately 10 mPa.s, it means relatively low. The question is how accurate is the measurement carried out with a FLUDICAM RHEO microfluidic visual rheometer. The sentence (l. 130-132) should be completed with the measured range as for lower values of shear rate the character of all curves would not be probably Newtonian. Newtonian character is closely connected with the region where hydrodynamic forces dominate.

·         It is not fully clear from the text how the individual components are involved in the solutions. It should be clearly indicated whether 10, 20, 30, 40 % are from the dye aqueous solution (l. 188) or from the pure water.

·         Fig. 5 - for better legibility the range of vertical axes should be sufficient between 50 and 80.

·         The Introduction and the Conclusion are practically identical.

·         The authors should formulate in a better way their aims and results.

 Just for improvement:

l. 68 - μS/cm

l. 83 - shear

l. 91- to separate a cipher and a unit

l. 111 - all

Table 2 - no dashes

l. 157 - decided ?

l. 158 - is ?

l. 18 + l. 199 - can: superfluous

Author Response

Dear Reviewer:

Thank you for the comments of our manuscript (Manuscript ID: polymers-471455). These comments are all valuable and very helpful for revising and improving our paper. We have made corrections according to the comments. The revised parts were marked in red characters. The responses to the comments are as follows:

Comment 1: To analyse the studied problem, it is necessary to measure some quantities which determination is not generally easy. Fig. 4 can serve as such example. For better visual evaluation of the measured data an additional vertical axis on the right side would be really beneficial. The lower line should represent water viscosity at 25 ℃. According to the tabulated data this value should be 0.89 mPa.s. However, a value depicted here seems to be by at least 50 % higher. On the other hand, viscosity at 25 ℃ for pure PEG 400 is according to the literature about 88 mPa.s. A value in Fig.4 for PEG400-40% is approximately 10 mPa.s, it means relatively low. The question is how accurate is the measurement carried out with a FLUDICAM RHEO microfluidic visual rheometer. The sentence (l. 130-132) should be completed with the measured range as for lower values of shear rate the character of all curves would not be probably Newtonian. Newtonian character is closely connected with the region where hydrodynamic forces dominate.

Response: Thank you, we have revised the part according to your comments. We have added vertical axis on the right side of corresponding figures (figure 2, line111; figure 4, line 149; figure 5, line 159). In Fig.4 (line 149), the lower line does not represent pure water, it is a dye aqueous solution contains 7% dye by weight, and the presence of dye molecules resulting in a higher viscosity. The solutions we tested is dye aqueous solution containing different concentration (0%, 10%, 20%, 30%, and 40% by weight) of PEG, so it has a lower concentration and viscosity and the preparation method of dye aqueous solution is redescribed in this paper (line 77). The viscosity at lower values of shear rate has been tested (figure 4a, line 149).

Comment 2: It is not fully clear from the text how the individual components are involved in the solutions. It should be clearly indicated whether 10, 20, 30, 40 % are from the dye aqueous solution (l. 188) or from the pure water

Response: Thank you, we have revised the part according to your comments. The concentration of 10, 20, 30, 40 % are from the PEG-dye aqueous solution. We redescribe the preparation method of dye aqueous solution in detail (line 77) and marked in red in the paper.

Comment 3: Fig. 5 - for better legibility the range of vertical axes should be sufficient between 50 and 80.

Response: Thank you, we have revised the part according to your comments. According to your suggestion, we have made a new figure (line 159).

Comment 4: The Introduction and the Conclusion are practically identical. 

Response: Thank you, we have revised the part according to your comments and marked in red in the paper.

Comment 5: The authors should formulate in a better way their aims and results.

Response: Thank you, we have revised the part according to your comments and marked in red in the paper.

Comment 6: Just for improvement: l. 68 - μS/cm, l. 83 – shear, l. 91- to separate a cipher and a unit, l. 111 – all, Table 2 - no dashes, l. 157 - decided? l. 158 - is? l. 18 + l. 199 - can: superfluous.

Response: Thank you, we have revised the part according to your comments and marked in red in the paper.

Reviewer 2 Report

The manuscript describes the jetting performance of the ink. The work looks interesting and minor revision is needed.

1# In Introduction, several sentences should be given to describe the importance of polyethylene glycols in the dye ink or why does polyethylene glycols discuss here?

2# In Introduction, authors mentioned Ohnesorge number. It is better for authors to provide one Table of their own ink in 3. Results and Discussion part.

3# To make the manuscript more clear, it is better for authors to provide the printing results on substrate, e.g. textile, as a demonstration using the best ink. 

4# For the manuscript, there should be a space between number and unit. 

Author Response

Dear Reviewer:

Thank you for the comments of our manuscript (Manuscript ID: polymers-471455). These comments are all valuable and very helpful for revising and improving our paper. We have made corrections according to the comments. The revised parts were marked in red characters. The responses to the comments are as follows:

Comment 1: In Introduction, several sentences should be given to describe the importance of polyethylene glycols in the dye ink or why does polyethylene glycols discuss here?

Response: Thank you, we have revised the part according to your comments. The importance of polyethylene glycols in the dye ink or why does polyethylene glycols have been discussed in the introduction and marked in red in the paper (line 61).

Comment 2: In Introduction, authors mentioned Ohnesorge number. It is better for authors to provide one Table of their own ink in 3. Results and Discussion part.

Response: Thank you, we have revised the part according to your comments. The inverse (Z) of the Ohnesorge number has been present in Table 2 (line 180) according your suggestion. 

Comment 3: To make the manuscript more clear, it is better for authors to provide the printing results on substrate, e.g. textile, as a demonstration using the best ink.

Response: Thank you, we have revised the part according to your comments. The printed cotton fabrics with high jetting performance dye solutions have been present in figure 11 (line 226) according your suggestion.

Comment 4: For the manuscript, there should be a space between number and unit.

Response: Thank you, we have revised the part according to your comments and marked in red in the paper.

Round  2

Reviewer 1 Report

The requirements were met.